# Biopsychosocial Predictors of Postpartum Depression: Protocol for Systematic Review and Meta-Analysis

**DOI:** 10.3390/healthcare12060650

**Published:** 2024-03-14

**Authors:** Marwa Alhaj Ahmad, Shamsa Al Awar, Gehan Sayed Sallam, Meera Alkaabi, Darya Smetanina, Yauhen Statsenko, Kornelia Zaręba

**Affiliations:** 1Institute of Public Health, College of Medicine and Health Sciences, United Arab Emirates University, Al Ain P.O. Box 15551, United Arab Emirates; 2Department of Obstetrics & Gynecology, College of Medicine and Health Sciences, United Arab Emirates University, Al Ain P.O. Box 15551, United Arab Emirates; sawar@uaeu.ac.ae (S.A.A.); gsayed@uaeu.ac.ae (G.S.S.);; 3Department of Radiology, College of Medicine and Health Sciences, United Arab Emirates University, Al Ain P.O. Box 15551, United Arab Emirates; daryasm@uaeu.ac.ae (D.S.); e.a.statsenko@uaeu.ac.ae (Y.S.); 4ASPIRE Precision Medicine Institute in Abu Dhabi, Imaging Platform, United Arab Emirates University, Al Ain P.O. Box 15551, United Arab Emirates

**Keywords:** postpartum, depression postpartum, blues postpartum, psychosis review article, risk factors

## Abstract

During the postpartum period, psychological disorders may emerge. Aims and objectives: With the current study, we aim to explore the biological determinants that act on women during labor and incur the risk for postpartum depression (PPD). To reach the aim, we will perform the following tasks: (i) identify biological peripartum risk factors and calculate pooled prevalence of PPD for each of them; (ii) explore the strength of the relationship between peripartum risk factors and PPD; (iii) rank the predictors by their prevalence and magnitude of association with PPD. The knowledge obtained will support the development and implementation of early diagnostic and preventive strategies. Methods and analysis: We will systematically go through peer-reviewed publications available in the PubMed search engine and online databases: Scopus, Web of Science, EMBASE. The scope of the review will include articles published any time in English, Arabic, or Polish. We will deduplicate literature sources with the Covidence software, evaluate heterogeneity between the study results, and critically assess credibility of selected articles with the Joanna Briggs Institute’s bias evaluation tool. The information to extract is the incidence rate, prevalence, and odds ratio between each risk factor and PPD. A comprehensive analysis of the extracted data will allow us to achieve the objectives. The study findings will contribute to risk stratification and more effective management of PPD in women.

## 1. Introduction

Postpartum depression (PPD) is a debilitating mental disorder. In a recent systematic review, the pooled prevalence of PPD among women after childbirth was 14.0% [1]. Determinants of PPD can be grouped into socioeconomic, demographic, and biological ones. The first group includes income, educational level, cultural background; the second group covers age, sex, race/ethnicity, marital status; the last group can be categorised into obstetric, maternal, and neonatal risks. With an exception to the mode of delivery, obstetric and other biological peripartum determinants of PPD are sparsely represented in the literature [2].

The prevalence of PPD varies significantly across countries with the highest numbers in South Africa and Southern Asia: 39.96 and 22.32%, respectively [1,3,4,5,6,7]. The lowest rates for PPD are documented in Oceania—11.11% [3]. In the Middle East, PPD affects 27% of mothers, and the United Arab Emirates is a country with an enormously high percentage of women suffering from this disorder—35% [6]. The high incidence of PPD in economically developed countries suggests the necessity of studying biological obstetric risk factors of the disease.

PPD significantly decreases quality of life [8,9,10]. Mothers with PPD struggle to perform daily chores, care for children, and establish a bond with them, which may negatively affect infant development [11,12,13]. Severe episodes of PPD can lead to infanticide [14]. The disorder may trigger dissatisfaction with marriage, paternal postpartum depression, violence, and divorce [15,16,17]. Untreated cases can develop into major depressive disorder and increase the risk of suicide [18,19,20,21,22,23]. The early detection and treatment of PPD help to maintain positive family dynamics after childbirth. However, risk stratification remains a challenge.

Recent studies on the prevalence of PPD and associated factors focused mainly on demographic and socioeconomic determinants [24,25,26,27]. According to the studies, mothers from all socioeconomic strata are at risk for depression [28]. Little is known about obstetric and other biological predictors of PPD, although some of the risk factors can be modified or used for screening purposes [29,30,31,32,33]. This serves as a motivation for the current study.

Since PPD is significantly undertreated, authors argue for the necessity of the preventative strategies that form more effective parenting skills and increased attachment to infants [34,35]. The occupational therapist maintains women’s readiness for childbirth. The specialist helps the patients to look at their limitations as at the problem which the women can solve [36,37]. Although efficient, the preventive measures are resource-consuming; therefore, they can be prescribed only to carefully selected cohorts of women who are at risk of this disease. To detect indications for PPD prevention, physicians need a reliable screening program with accurate risk assessment.

The existing screening for PPD does not meet the demand of time. First, some women may be reluctant to share symptoms: they worry about how inefficiency in parental responsibilities is perceived by other people and want to protect personal social standing [38]. Second, a routine assessment of the mothers’ mood is not consistently performed in some healthcare institutions. Third, biological, psychological, and socioeconomic determinants act simultaneously [39,40,41,42,43]. In this meta-analysis, we will explore relationships between PPD and multiple biological risk factors in the peripartum period. These factors can be detected directly from electronic patients’ data. The systematic synthesis of these data will promote evidence-based practices of PPD screening, prevention, and treatment [44].

## 2. Objectives

The meta-analysis aims to explore the biological determinants that act on women during labor and incur the risk for PPD. The objectives of this project are as follows:Identify biological peripartum risk factors and calculate pooled prevalence of PPD for each of them.Explore associations between biological peripartum risk factors and PPD.Rank the predictors by their prevalence and strength.

## 3. Materials and Methods

We will follow the checklist of the Preferred Reporting Items for Systematic Review and Meta-Analysis Protocol (PRISMA-P) [45]. The PRISMA-P checklist is available in online Appendix A. The protocol is registered with the international database for systematic reviews PROSPERO (registration number CRD42022372067).

### 3.1. Study Design and Data Source

To perform a comprehensive systematic literature review, we will submit queries to three biomedical databases (Web of Science, EMBASE, and Scopus) and PubMed search engine. The search keywords and medical subject headings are listed in Table 1. We will extract English, Arabic, or Polish papers without time restrictions and screen the retrieved papers manually.

### 3.2. Eligibility Criteria

The review will cover generally healthy females without known risks for PPD before the last childbirth. Fetal abnormalities, maternal diseases, violence, and other traumatic life experiences will serve as exclusion criteria. To avoid bias connected with history of the risk factor, we will focus on the last childbirth, its complications, and birth modality. We will consider for the review only the original publication reporting findings on the women who live healthy lifestyles.

This study will analyse peer-reviewed papers about changes in postpartum mood, PPD, depressive disorder, or suicidal ideation in females after the delivery without restrictions on their age. The review scope will also include publications about pregnant women who had psychiatric consultation/referral or suicidal attempts following childbirth. Table 2 summarises the inclusion and exclusion criteria for the literature.

We will exclude grey literature, protocol papers, editorial letters, reviews, and case studies from the current review. Articles describing mental problems, neurocognitive diseases, and mood disorders prior to the delivery will not be considered. Also, the project will not cover COVID-19-related factors of postpartum mental health in women.

### 3.3. Study Records

Selection process: papers extracted from the biomedical databases will be uploaded to the Covidence software (https://www.covidence.org/) for automatic deduplication and further screening. The initial title and abstract screening will be performed against the eligibility criteria by two reviewers. Then, the whole text will be examined. If the reviewers disagree on eligibility of an article, the third reviewer will resolve the conflict. The PRISMA flowchart will show the selection process and results.

Data extraction: two reviewers will independently extract the following information to an online spreadsheet: authors, country, publication year, sample size, study design, and data related to the PPD assessment and biological risks. Table 3 presents the full list of target variables and possible confounders. For each determinant, we will collect data on PPD prevalence, scores, and applied cut-off values. We will consider the following assessment tools: Edinburgh Postnatal Depression Scale, Beck Depression Inventory, and Patient Health Questionnaire-9.

From studies with the cross-sectional design, we will extract effects’ estimate expressed as the odds ratio (OR) or risk ratio with corresponding measures of precision (*p*-value, confidence interval). From observational studies, the following data will be collected: the number of women with and without PPD in the observational and control groups. The extracted findings will be used to calculate the pooled OR and risk ratio. In the comprehensive analysis, socioeconomic and demographic determinants will also be extracted as possible covariates: one can use this information for calculating adjusted OR.

For analyzing the relationship between two continuous variables over numerous studies, we will perform meta-correlation. The reviewers will derive the study size, r-values, and 95% confidence interval. These correlation coefficients will be combined using meta-analytic techniques to obtain a general approximation of the strength of association. The principal investigator will contact correspondent authors for any missing data.

Quality assessment of individual studies: we will use Joanna Briggs Institute checklists for a critical appraisal of cross-sectional, case–control, and cohort studies [46]. Two reviewers will independently assess each study against the corresponding checklist. In case of their disagreement, the third reviewer will decide on the final quality score of the study. The research team will apply funnel-plot-based methods to deal with potential publication bias [47]. In particular, we will construct funnel plots with Begg’s and Egger’s test [48,49].

Data analysis and synthesis: once the data extraction is completed, we will check articles for inter-study homogeneity with the I2 test [50]. The possible sources of heterogeneity are the age of study participants and the time passed from childbirth. Study cohorts may also differ in the formal diagnosis of PPD and its severity. To control for this difference, we will divide the total population into two categories: the women at risk of the disorder and the women diagnosed with PPD. We will use weighted prevalence functions to continue the meta-analysis even if the I2 index exceeds 75%, which indicates a strong between-study variability [51].

The subgroup analysis will be performed to evaluate the consistency of findings across multiple observational groups. In this way, we will deal with the anticipated high inter-study variability. Specifically, we plan to apply a random effects model, which helps to generalise findings beyond the included papers [52]. All articles will be grouped according to the peripartum risk factors that they describe: birth management (mode of birth, epidural anesthesia), maternal complications (hemorrhage, vaginal lacerations, etc.), neonate complications (APGAR score, shoulder dystocia, diseases, etc.).

### 3.4. Study Methodology

To address the first specific objective, the research team will look for the biological peripartum determinants of PPD. In this study, we will consider the factors that have biologic nature, act at the time of delivery or shortly after it, and pose a risk to the women’s mood postpartum. Then, we will calculate the pooled incidence and prevalence of PPD in women for a specific group of peripartum complications.

Working on the second specific objective, we will explore the relationship between PPD and peripartum complications. An additional subgroup analysis will be conducted to identify the relationship between peripartum risk factors and the severity of PPD according to Edinburgh Postnatal Depression Scale, Beck Depression Inventory, and other questionnaires. The link between variables will be expressed in the OR, r-coefficient, and *p*-values.

For the third specific objective, we will create ranking charts representing the contribution of different etiological factors to the total PPD incidence (see Figure 1). The statistical analysis will be performed in R package “meta” [51]. After constructing forest plots and ranking charts, we will perform sensitivity analysis with the leave-one-out method to check the robustness of the final results and to assess the effect of a single study on the overall outcome.

## 4. Discussion

### 4.1. Pathogenesis of Postpartum Depression

The proposed meta-analysis will summarise existing knowledge on the role of biologic peripartum determinants of mental health in women after childbirth. Herein, we analyse pathogenic mechanisms of PPD to identify and categorise the risks that have biologic nature as opposed to social, economic, and demographic determinants. The major risks fall under the umbrella of stress and adaptation to a new environment [53].

Persistent negative thoughts, anxiety, and depression may result from a negative birth experience related to obstetric complications and neonate pathologies [54]. Neuroendocrine abnormalities, such as elevated levels of stress hormones, underlie the biology of postpartum mood disorders. Thyroid hormones have also been implicated in PPD: the known risk factors are the overt thyroid dysfunction and mere presence of thyroid antibodies even during early pregnancy [55,56]. However, literature findings are inconclusive, and the reports on the thyroid-stimulating hormone in PPD are even more ambiguous [57].

Some studies suggest sleep quality but not hormones as a predictor of the time of PPD recurrence [58,59]. However, a challenging question is whether the sleep disorder is the cause or the consequence of depression [56].

Inflammatory response is another mechanism of PPD development. In the normal labor and delivery, the response is moderate, but its exaggeration might lead to PPD as a psycho-neuro-immunological disorder [60]. Infection, injury, and stress trigger the innate immune system, and the elevated levels of proinflammatory cytokines pose a risk to women through the identified mechanisms that all have inflammatory nature: stress, sleep disturbances, pain, inflammation, psychological trauma and history of depression or trauma, etc. [60,61]. However, peripartum or postpartum levels of the cytokines are not predictive of later development of PPD [59,62].

### 4.2. Biological Risks of Postpartum Depression

Biological determinants of PPD can be categorised into maternal, neonate, and birth-management-related risks (see Table 3). Information on these subgroups is scarcely presented in the literature. Furthermore, observational studies produced conflicting findings about the role of labor complications in the development of PPD. The adverse psychological effect of neonatal pathology has not been studied well [63].

Birth-management-related risks include peripartum emergencies, postpartum complications, types of delivery, anaesthesia, etc. [64,65]. The type of birth is the only biological determinant extensively covered in recent studies: the cesarean section (CS) is a risk of PPD due to fear, preoperative anxiety [66], post-traumatic stress [67], and general anesthesia [68]. However, the vaginal delivery may also induce PPD due to postpartum hemorrhage [69,70], genital lacerations [71,72], etc. The personal preference in the delivery mode may impact the individual perception of the childbirth experience and postpartum health [73].

An incomplete list of obstetric complications include protracted cervical dilation, endometriosis, preeclampsia, gestational diabetes, miscarriage, and preterm childbirth in the future pregnancies [74,75,76,77,78,79,80,81]. Some studies did not find a direct association between the postpartum hemorrhage and PPD [82,83]. Still, post-hemorrhagic anemia, negative birth experience, and fear of death increase the risk of maternal depression [84,85,86,87].

In this research, we will review the obstetric procedures that were not properly studied as risk factors of PPD. For example, episiotomy incision may cause bleeding, swelling, infection, and perineal pain that affect quality of life, sexual and mental health in mothers [84,88,89,90].

Postpartum uterine curettage is associated with bleeding, anemia, and psychological discomfort [84,85,86,87]. The curettage may damage the cervix, trigger Asherman’s syndrome, cause infection, cause dys- and oligomenorrhoea [91]. Adverse effects of obstetric procedures include restrictions on sexual life and physical activity, thus leading to mood disorders [92].

Neonatal status refers to such characteristics as the birth weight, gestational age, APGAR score, health complications, admission to the neonatal intensive care unit [87,93,94,95,96,97]. Severe conditions, such as asphyxia, can significantly impair mother’s mental health. In addition, the neonate’s temperament [93], sleep patterns [98], and breastfeeding behavior can also be stressful [99], which leads to anxiety and PPD.

Maternal risks. If developed before the childbirth, maternal pathology will serve as the exclusion criterion for literature selection in the proposed study since psychological vulnerability of the ladies is considerably increased in these conditions. For example, neurological and heart conditions double the risk of PPD with OR of 2.37 and 1.92, respectively [100]. Migraine is also a strong determinant of peripartum mental illnesses [101]. Maternal pathology may induce mood disorders by raising the risk of obstetric complications [102,103].

### 4.3. Socioeconomic and Demographic Confounders

The role of socioeconomic and demographic determinants in developing PPD is widely covered in the literature. However, some findings remain controversial: a recent systematic review showed a U-shaped form of relationship between PPD prevalence and maternal age. From 24 to 35 years of age, women are less prone to PPD [40]. A prospective study revealed a similar pattern [104]. Contrarily, other authors did not find a prominent difference in odds among age groups [105,106]. In addition, the increased maternal age can be considered a biological risk due to a close link with the obstetric complications that modulate the psychological status of females.

Racial and ethnic disparities in PPD are also considerable but data on these factors should be interpreted with caution for the following reasons: First, the patient’s self-reported race may not correlate with the genetic ancestry [107]. Second, ethnicity is rather the environmental determinant than the biological one because of a close association with the language [107]. Third, racial bias could play a vital role in overall well-being. A study showed that odds of PPD are higher for women who experience racial bias [108].

Social/wealth inequity impacts psychological health. A recent study reported greater rates of PPD in the countries with a higher income disparity [109]. In the non-uniform health insurance scheme, low economic status is stressful due to the inaccessibility of healthcare. However, in Denmark, PPD is more common in low-earning families despite the free universal healthcare system [82]. A high prevalence of PPD is also typical for the countries where young women work more than 40 hours per week [109]. Socioeconomic factors are not specific for PPD, and they can increase the likelihood of other diseases [110].

## 5. Conclusions

A high global prevalence of PPD calls for an efficient strategy for screening and prevention. The disease stems from multiple psychological, familial, social, and cultural factors that should be studied in combination to develop an evidence-based approach to PPD prevention. Still, little is known about the obstetric and other biological determinants. Hormonal imbalance and immune alterations are the major pathophysiological mechanisms of developing PPD which it can be triggered by peripartum emergencies, postpartum complications, mode of birth, obstetric procedures, and neonatal status. Physicians should also consider individual beliefs, cultural and social norms through the prism of which the woman appraises childbirth. In the proposed review, we will primarily focus on biological peripartum risks and adjust them for psychosocial confounders.

## 6. Strength and Limitations

The protocol is prepared in accordance with the PRISMA-P checklist for systematic reviews; the protocol is registered with the international database for systematic reviews PROSPERO.The study will focus on the biological peripartum risk factors for PPD, which are not studied well.The socioeconomic and demographic risks will also be included in the analysis as confounders.We will perform subgroup analysis to evaluate the consistency of findings across multiple observational groups. If the risk of PPD varies markedly among different categories of patients, we will check whether studies replicate and add confidence to the findings. If not, we will not derive conclusions from such data.A notable limitation of the proposed study is scarcity of findings on biological peripartum risk factors for PPD. If we fail to perform the meta-analysis, a narrative systematic review will be conducted.

## Figures and Tables

**Figure 1 healthcare-12-00650-f001:**
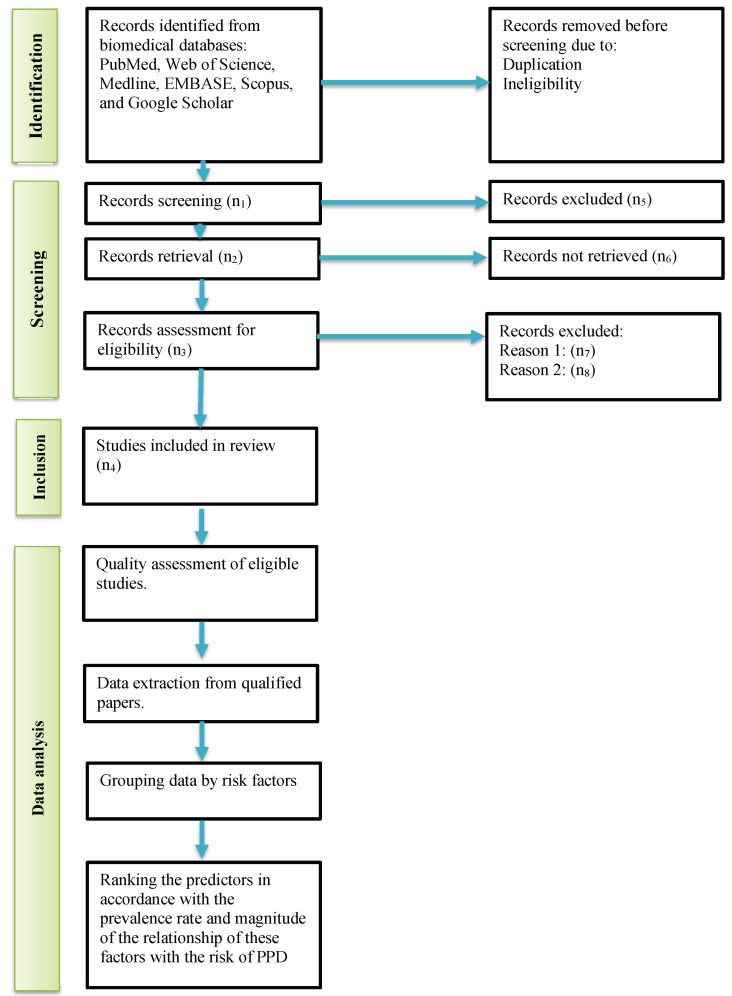
Study pipeline.

**Table 1 healthcare-12-00650-t001:** Keywords and medical subject headings for PubMed/Medline.

No.	Search String	Number of Articles
1	“postpartum period”[MeSH Terms] OR “postpartum”[Title/Abstract] OR “puerperium”[Title/Abstract] OR “pregnancy”[MeSH Terms] OR “pregnancy”[Title/Abstract]	1,136,113
2	“depression, postpartum”[MeSH Terms] OR “depressive disorder”[MeSH Terms] OR “depression”[MeSH Terms] OR “depressive disorder”[MeSH Terms] OR “mood disorders”[MeSH Terms] OR “suicide”[MeSH Terms] OR “postpartum depression”[Title/Abstract] OR “mood disorder”[Title/Abstract] OR “baby blues”[Title/Abstract]	361,517
3	((((((((((((((((((((((risk factors[MeSH Terms]) (Obstetric Labor Complications[MeSH Terms])) OR (fetal disease[MeSH Terms])) OR (pregnancy complications[MeSH Terms])) OR (complication*[Title/Abstract])) OR (intrapartum complication*[Title/Abstract])) OR (birt complication*[Title/Abstract])) OR (maternal complication*[Title/Abstract])) OR (shoulder dystocia[Title/Abstract])) OR (hemorrhage[Title/Abstract])) OR (hemorrhage[Title/Abstract])) OR (asphyxia[Title/Abstract])) OR (baby complication[Title/Abstract])) OR (vaginal birth[Title/Abstract])) OR (vaginal delivery[Title/Abstract])) OR (caesarean section[Title/Abstract])) OR (vacuum extractor[Title/Abstract])) OR (forceps delivery[Title/Abstract])) OR (vaginal tears[Title/Abstract])) OR (vaginal laceration[Title/Abstract])) OR (episiotomy[Title/Abstract])) OR (uterine curettage[Title/Abstract])	2,579,080
4	“Forecasting”[MeSH Terms:noexp] OR “predict*”[Title/Abstract] OR “determinants”[Title/Abstract]	2,264,321
5	String #1 AND String #2 AND String #3 AND String #4	2156

* the asterisk sign is a universal symbol for truncation, i.e., the keyword spelling may vary in the search.

**Table 2 healthcare-12-00650-t002:** Inclusion and exclusion criteria.

Inclusion Criteria	Exclusion Criteria	
	**For Literature**	**For Participants**
1. Cross-sectional or longitudinal design original studies. 2. English, Arabic, or Polish peer-reviewed articles. 3. Articles reporting risk factors for PPD. 4. Studies focusing on changes in postpartum mood, suicidal ideation, and suicides following the last birth. 5. Study subjects who had psychiatric consultation or referral due to symptoms of depression.	1. Grey literature. 2. Case studies, reviews, meta-analyses, and letters to the editor case studies. 3. Research describing mental problems, neurocognitive diseases, and mood disorders that were present before birth. 4. Studies that did not report sensitivity and specificity. 5. Studies reporting the relationship between PPD and COVID-19-associated factors.	Pregnant women with the following diseases and conditions diagnosed before the last birth: 1. Mental and psychological disorders (F00–F99 in ICD-10). 2. Cerebrovascular diseases (I60–I69). 3. Organic pathologies of the central nervous system (e.g., brain and meninges tumours—C71, D32–33). 4. Serious abnormalities or diseases known before the last birth that are known risk factors for PPD (O35.9 in ICD10). 5. Partner or other type of violence.

**Table 3 healthcare-12-00650-t003:** Determinants of postpartum depression: target variables and confounders.

Group	Subgroup	Variables
Obstetric determinants	Birth-management-related risks	Type of delivery:
		- Spontaneous vaginal delivery
		- Assisted vaginal delivery (forceps or vacuum extraction)
		- Planned (elective) caesarean section
		- Emergency caesarean section
		Type of anaesthesia:
		- Epidural
		- Spinal
		- Local (pudendal block)
		- General
		Fetal lie/presentation/position:
		- Shoulder dystocia
		Peripartum emergencies:
		- Amniotic fluid embolism
		- Intrapartum/postpartum hemorrhage
		- Pulmonary embolism
		- Placenta abruption
		- Uterine rupture
		- Fetal bradycardia
		- Severe perineal tears (III and IV degree)
		- Severe vaginal lacerations
		- Prolonged I, II, or III stage of labour
		Assistance in labour:
		- Induction of labour
		- Episiotomy
		- Amniotomy
		Drugs in delivery:
		- Oxytocic hormones
		- Prostaglandin E1 analogues
	Maternal risks	Complications:
		- Postpartum anemia
		- Postpartum endometritis
		- Urinary incontinence
		Others:
		- Length of stay in hospital after delivery
		- Time since giving birth
	Neonate risks	APGAR score
		Hypoxia
		Abnormalities found after delivery
Confounders	Demographic risks	Age group
		Country of study
		Race/ethnicity
	Socioeconomic risks	Socioeconomic status (low, medium, high)
		Level of education
		Income
		Marital status
		Employment status

## Data Availability

Data are contained within the article and Appendix A.

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
