# Peer review of "Biopsychosocial Predictors of Postpartum Depression: Protocol for Systematic Review and Meta-Analysis"

_healthcare, 2024, doi:10.3390/healthcare12060650_

Round 1

Reviewer 1 Report

Comments and Suggestions for Authors

Dear Authors,

Thank you so much for the commendable effort in putting this article together.

I have gone through it and I feel that this is a well-thought-out piece of systematic review. However, I have a few comments that you may consider to make this article better.

Specific Comments:

-The article is well written and the Preferred Reporting Items for Systematic Reviews and Meta-Analyses (PRISMA) have been considered in most of the steps.

-In the abstract, it is clear that the authors are interested in raising awareness among healthcare professionals about postpartum depression in women. is should be seen as a running thread in the systematic review Moreover, the title can read "Biopsychosocial predictors of Postpartum Depression: Protocol for Systematic Review and Meta-Analysis.

-In line in the introduction, is 14% a global prevalence of mental disorders among women after birth?

-In line 33 there is a need for citation to qualify this statement....' PPD significantly decreases quality......."

- ln the strengths and limitations section, the authors should endeavour to explain how they will counter the shortcomings of this proposed protocol.

For instance "A notable limitation of this systematic review is the scarcity of findings on biological peripartum risk factors for PPD. How will you deal with this?

Finally, I wish you all the best and will be glad to read your final product sometimes in the near future.

Author Response

Dear Reviewer, 

kindly see the attached document for the detailed response. 

Thank you,

Authors of the manuscript

Reviewer 2 Report

Comments and Suggestions for Authors

Dear authors, the subject matter is very interesting and challenging, but your manuscript has several structural and functional errors that make it currently unpublishable, for the following reasons:

1) the manuscript has "meta-analysis" in the title but there is no data evaluation and statistics in the text, just as there is also "systematic review" in the title but the manuscript despite being registered since 2022 on PROSPERO the process of article selection is not clear and timely as required in the guidelines on the publication of systematic reviews. It is currently closer to a narrative and therefore authors have to make a choice. 

2) Figure 1 is insufficiently detailed and descriptive and leaves considerable doubt about the manuscript selection process; furthermore, there is no summary table of selected articles, should authors wish to lean toward the systematic line.

3) the discussions and conclusions are excessively sparse and schematic, and would deserve more detail based on the statistics to be carried out on the data (and which are lacking in their detail in the manuscript).

4) the date in PROSPERO is November 2022 while the manuscript has a date of June 2023 inserted. This is a discrepancy that, taking into account the limitations they have reported, raises the suspicion that the authors intended to publish original research after this thematic introduction, which, however, does not have the defined and complete characteristics to be systematic review and/or meta-analysis. 

For these reasons, I suggest that authors provide more clarity on the manuscript about:

a) the nature of the manuscript (and I suggest "narrative" or "systematic" if they include tables and improve the process of article selection, or again "meta" if however they perform all the necessary statistics and improve the selection process;

b) the complete and detailed list of variables under consideration, with specification both in the background/introduction and in the results, and thus in the discussions;

c) the missing statistics;

d) the missing tables and graphs;

e) the expansion of the discussions and conclusions. 

Author Response

(The authors gave the same response as above.)

Reviewer 3 Report

Comments and Suggestions for Authors

Healthcare-2875969

It is a protocol that is written in a confusing manner, referring to the lack of knowledge of the biological factors that are associated with the development of postpartum depression.

The authors maintain that to identify biological risk factors it will be necessary to analyze the pathogenic mechanisms of postpartum depression, as well as adaptive changes in neuropsychoendocrinology and alterations in neurotransmission and brain connectivity. However, in the material and methods section only reference is made to obtaining the variables related to the peripartum and postpartum, as well as some sociodemographic variables, which will not allow the proposed analysis of the pathogenic mechanisms of postpartum depression to be carried out.

The introduction is very confusing, it is suggested to present a theoretical framework that supports your research question, as well as present an appropriate approach to the problem.

There is no coherence between the proposed objective and the information that is supposed to be obtained after carrying out the protocol.

The authors carry out a discussion that has nothing to do with what was initially proposed.

The manuscript has paragraphs that are repetitive, such as the exclusion criteria section.

Author Response

(The authors gave the same response as above.)

Reviewer 4 Report

Comments and Suggestions for Authors

Dear authors, I congratulate you on a very well-designed study protocol for such an important and under-researched topic as PPD and biological risk factors for its development. It is particularly interesting that you also noticed that the incidence of PPD is high in developed countries where the trend is for women to give birth electively by caesarean section. I'm really looking forward to the results to eventually show if elective caesarean section carries with it a higher risk of developing PPD. I think it's a really important topic that will change the views of ordinary women about the safety aspects of caesarean section.

The study was designed at a high level and will provide valid results.

The only two things I would like you to clarify for me are why you chose to take studies only in Arabic, English and Polish? Isn't that discrimination and won't you get an invalid sample that way? Wasn't it better to stick to English only or take studies in Spanish, Italian, German, French, Chinese? That way you would have almost the entire planet covered.

And the second thing is that there are occasional typographical errors in the text that need to be corrected and edited.

With certain clarifications and corrections, I believe that this protocol is good and acceptable for publication in this journal. I thank you for your efforts and look forward to the results of the study. I wish you all the best.

Comments on the Quality of English Language

Only a few typos in the text. Apart from that, english language is fine and does not need corrections.

Author Response

(The authors gave the same response as above.)

Round 2

Reviewer 2 Report

Comments and Suggestions for Authors

I have noted the changes made and for me it can be published. Thank you

Author Response

Dear Reviewer,

Thank you for your supporting comments. 

Kind regards,

Co-authors

Reviewer 3 Report

Comments and Suggestions for Authors

Healthcare-2875969-peer-review

Why limit yourself only to Scopus, Web of Science, EMBASE databases and only documents in English, Arabic or Polish?

When the authors propose a protocol that researchers can use to determine the different biopsychosocial predictors of postpartum depression in different regions of the world.

Why restrict the researcher?

In the introduction, reference must be made to the few studies published on the identification of biological determinants and how they have influenced postpartum depression.

If the publication of articles referring to biological determinants is non-existent or very scarce, carrying out a meta-analysis is not a strategy to obtain information on these determinants, but rather the proposal should be to carry out cross-sectional, case-control or cohort studies to have enough information, with which a systematic review and meta-analysis can be carried out with the purpose of having valid conclusions, especially when there are discrepancies or there is no consensus. By having this information from the systematic review, preventive strategies can be carried out. If your argument is that there is no research on this topic, what will be analyzed in a systematic review?

Author Response

Dear Reviewer,

kindly see the reply in the attached Word file.

Kind regards,

Co-authors
